# ZebraLogic: On the Scaling Limits of LLMs for Logical Reasoning

Bill Yuchen Lin [1]      Ronan Le Bras [2]

Kyle Richardson [2]     Ashish Sabharwal [2]     Radha Poovendran [1]     Peter Clark [2]     Yejin Choi [3]

[1] University of Washington      [2] Allen Institute for AI      [3] Stanford University

byuchen@uw.edu    ronanlb@allenai.org    yejinc@stanford.edu

https://hf.co/spaces/WildEval/ZebraLogic

## Abstract

We investigate the logical reasoning capabilities of large language models (LLMs) and their scalability in complex non-monotonic reasoning. To this end, we introduce ZebraLogic, a comprehensive evaluation framework for assessing LLM reasoning performance on logic grid puzzles derived from constraint satisfaction problems (CSPs). ZebraLogic enables the generation of puzzles with controllable and quantifiable complexity, facilitating a systematic study of the scaling limits of models such as Llama, o1 models, and DeepSeek-R1. By encompassing a broad range of search space complexities and diverse logical constraints, ZebraLogic provides a structured environment to evaluate reasoning under increasing difficulty. Our results reveal a significant decline in accuracy as problem complexity grows—a phenomenon we term the "curse of complexity." This limitation persists even with larger models and increased inference-time computation, suggesting inherent constraints in current LLM reasoning capabilities. Additionally, we explore strategies to enhance logical reasoning, including Best-of-N sampling, backtracking mechanisms, and self-verification prompts. Our findings offer critical insights into the scalability of LLM reasoning, highlight fundamental limitations, and outline potential directions for improvement.

## 1. Introduction

Logical reasoning stands as a cornerstone of human intelligence and remains a central challenge in AI. While recent advances have demonstrated promise in tasks requiring common sense and general knowledge (Brown et al., 2020; Chowdhery et al., 2022; Bubeck et al., 2023), the capabilities of Large Language Models (LLMs) in handling complex deductive problems remain uncertain. This limitation in our understanding is especially critical as systematic reasoning underpins many real-world applications. To systematically study LLMs' logical reasoning capabilities and their scaling limits, an ideal evaluation framework must: (1) isolate pure logical reasoning from domain knowledge; (2) enable precise control over problem complexity; (3) minimize data leakage to prevent training data memorization; (4) provide objective metrics for assessing an LLM's reasoning results.

Constraint satisfaction problems (CSPs) offer such a controlled framework (Dechter, 2003): they are mathematically well-defined, scalable in both complexity and search space, and have solutions that can be automatically verified. By formulating logical tasks as CSPs, we can rigorously evaluate how well LLMs adhere to logical constraints, independent of domain-specific data or heavy numerical computation. As a representative class of CSPs, logic grid puzzles (specifically Zebra Puzzles or Einstein's Riddle, (Prosser, 1993)) are particularly suitable as they require pure formal reasoning, remain accessible enough to serve as an effective testbed, and embody core skills relevant to real-world applications such as task planning, scheduling, and resource allocation. Hence, we introduce **ZebraLogic**, an evaluation framework for creating logic puzzles with controllable, and quantifiable complexity, thus improving our understanding on the scaling limits of LLMs including Llama (AI@Meta, 2024), o1 (OpenAI, 2024) and R1 (DeepSeek-AI, 2025). [1]

Through extensive evaluation of various LLMs across di-

---

[1]University of Washington [2]Allen Institute for AI [3]Stanford University. Correspondence to: Bill Yuchen Lin <byuchen@uw.edu>.

*Proceedings of the 42nd International Conference on Machine Learning*, Vancouver, Canada. PMLR 267, 2025. Copyright 2025 by the author(s).

---

[1]The closest related effort is by Tyagi et al. (2024), who focused on a detailed analysis of the kinds of errors LLMs make when solving grid puzzles using reasoning chains; more details in §7.

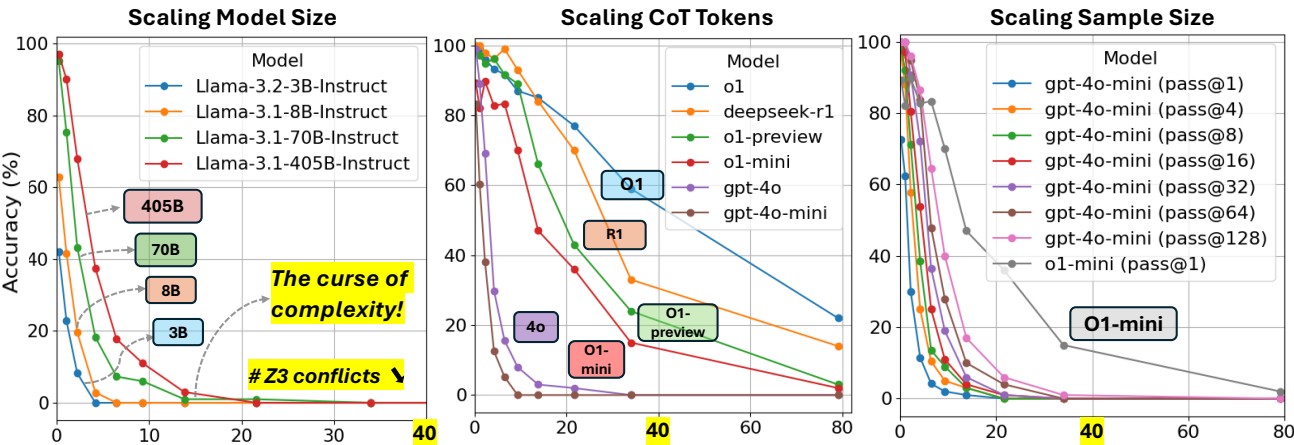

Figure 1: Accuracy vs number of Z3 conflicts for Llama-3 (left), showing the size scaling effect on the reasoning performance. The middle figure shows the curves for gpt-4o(-mini) vs o1 and R1, showing the scaling effect of model size and test-time compute. The right figure shows the scaling effect of repeated sampling by pass@k metric with different sample sizes.

verse architectures and sizes, we observe a dramatic decline in performance as puzzle complexity increases—a phenomenon we term the "**curse of complexity for reasoning**." Most models struggle once the puzzle's search space exceeds $10^7$ possibilities (e.g., for puzzles with 4x5 grid size) or when the number of logical conflicts in a widely used SMT solver named Z3 (de Moura & Bjørner, 2008) surpasses 20. These findings suggest that limited reasoning in current LLMs are not solely a matter of model- or sample-size scaling, but also arise from insufficient test-time compute. This shortfall underscores the need to train LLMs to reason step by step (Wei et al., 2022) explicitly (e.g., via reinforcement learning (Lambert et al., 2024a)), as exemplified by emerging reasoning models such as o1 and R1. Specifically, we conduct a systematic investigation into the scaling behavior of LLMs in logical reasoning, focusing on three key dimensions: model size (§4), sampling (§5), and test-time compute (§6). Understanding scaling behavior of LLMs in reasoning is critical to identify the most promising directions for advancing LLMs' reasoning capabilities and to guide future research efforts more effectively.

Our work makes the following key contributions:

- We create the ZebraLogic dataset, a benchmark of 1,000 logic grid puzzles spanning multiple complexity levels, designed to evaluate LLMs' logical reasoning capabilities systematically with two complexity metrics: search space size and Z3 conflict count (§2).
- We report "the curse of complexity" in logical reasoning with LLMs: the performance dramatically declines as the problem complexity increases and after a certain threshold, most models struggle to solve any logical puzzle. This limitation persists even when scaling to significantly larger models (such as Llama-3.1-405B) or using enhanced training data, indicating a deeper challenge that

cannot be resolved by model scaling alone (§3 and §4).
- We scale the test-time compute of LLMs by increasing the number of generation samples, revealing that it has both promise and challenges. While Best-of-N sampling can improve potential performance, practical selection methods like majority voting or reward models show limited improvement. Additionally, even pass@128 cannot break the curse of complexity (§5).
- We find that it's much more promising to scale up the reasoning tokens (i.e., chain-of-thoughts; CoTs) generated during inference with a backtracking mechanism. We take OpenAI's o1 models as a typical example and show that they generate significantly more, nearly 10x (hidden) reasoning tokens than others, which scale properly with problem complexity. Based on our empirical results, we also find that there exists an optimal ratio of reasoning tokens to Z3 conflicts, but O1-like models cannot always reach this optimal ratio when the complexity is extremely high, thus not achieving perfect reasoning (§6).
- Moreover, we explore the potential of using self-verification prompting to improve LLMs (§6.2). We find that such methods can help LLMs improve their performance, but the improvement is very marginal. We further analyze the reasoning process of o1 and discuss its strengths and weakness in logical reasoning (§D).

## 2. Problem Formulation of Logical Reasoning

Constraint Satisfaction Problems (CSPs) provide a powerful framework for modeling and solving logical reasoning tasks. In CSPs, solutions must satisfy a set of constraints over variables and their possible values. This framework is particularly valuable for evaluating systematic reasoning capabilities, as it requires explicit handling of logical relationships and dependencies. We leverage this frame-

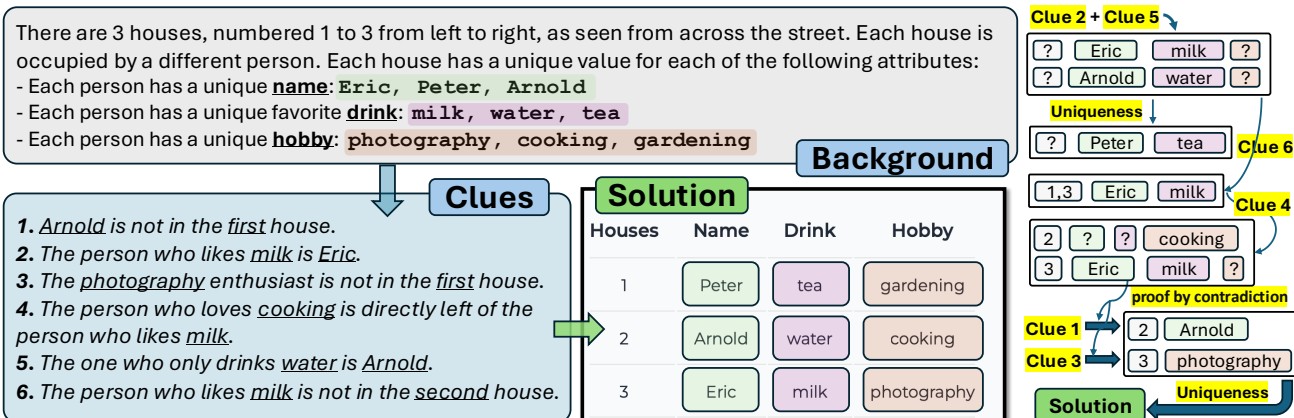

There are 3 houses, numbered 1 to 3 from left to right, as seen from across the street. Each house is occupied by a different person. Each house has a unique value for each of the following attributes:
- Each person has a unique **name**: `Eric, Peter, Arnold`
- Each person has a unique favorite **drink**: `milk, water, tea`
- Each person has a unique **hobby**: `photography, cooking, gardening`

**Clues**

*1. Arnold is not in the first house.*
*2. The person who likes milk is Eric.*
*3. The photography enthusiast is not in the first house.*
*4. The person who loves cooking is directly left of the person who likes milk.*
*5. The one who only drinks water is Arnold.*
*6. The person who likes milk is not in the second house.*

**Solution**

| Houses | Name | Drink | Hobby |
|--------|------|-------|-------|
| 1 | Peter | tea | gardening |
| 2 | Arnold | water | cooking |
| 3 | Eric | milk | photography |

Figure 2: This example of ZebraLogic features 3 houses (N=3) and 3 attributes (M=3), with 6 clues (K=6). The *Background* outlines the attributes, their possible values, and the uniqueness constraints. The *Clues* provide additional constraints regarding the *attributes*. The task for the model is to determine the correct assignment of attributes to each house based on these clues, as illustrated in the *Solution* grid.

work through logic grid puzzles in our ZebraLogic dataset to assess LLMs' deductive reasoning abilities.

### 2.1. Logic Grid Puzzles

Each puzzle in ZebraLogic consists of $N$ houses (numbered 1 to $N$ from left to right) and $M$ different attributes for each house. There are $N$ distinct values for each attribute, and each house must have a unique value for each attribute. Given a list of $K$ clues, one must use logical deduction to determine the unique correct assignment of values. Figure 2 illustrates an example of such a puzzle, as well as a reasoning chain for solving it. Importantly, while some ZebraLogic puzzles can be solved through straightforward linear deduction, many require more complex *non-monotonic* reasoning strategies, such as counterfactual reasoning that involves backtracking and revising assumptions. This is particularly true as the search space grows larger and the clues become more intricate – a key aspect of our study on the scaling behavior of LLMs.

### 2.2. Problem Formulation

We provide a detailed mathematical formulation of logic grid puzzles as a CSP. This formulation not only clarifies the underlying structure of the puzzles in ZebraLogic but also highlights how our study can be generalized to various reasoning problems. The example shown in Fig. 2 illustrates this formulation.

**Background.** Consider $N$ houses numbered 1 to $N$. Each house has a different occupant with a set $\mathcal{A}$ of $M$ unique attributes such as name, favorite drink, hobby, etc. Each attribute $a \in \mathcal{A}$ represents a category of characteristics and takes values in a set $\mathcal{V}_a$ of $N$ possible values. For

example, for the attribute `Name`, we might have $\mathcal{V}_{\text{Name}} = \{\text{Eric}, \text{Peter}, \text{Arnold}\}$ in a puzzle with $N = 3$ houses. As illustrated in Fig. 2, other attributes might include `Drink` with values like milk, water, and tea, or `Hobby` with values like photography, cooking, and gardening. To model the puzzle as a Constraint Satisfaction Problem, we define variables representing the assignment of values to attributes for each house.

- Let $H = \{1, 2, 3, \cdots\}$ be the set of houses, $|H| = N$.
- Let $\mathcal{A} = \{\text{Name}, \text{Drink}, \cdots\}$ be the set of attributes, $|\mathcal{A}| = M$.
- Define $x_{a,k} \in \mathcal{V}_a$ for each attribute $a \in \mathcal{A}$ and house $k \in H$.

**Uniqueness Constraints:** The constraints ensure that each value is assigned exactly once, as described in the Background part in Figure 2. For each attribute, the set of assigned values across all houses must exactly match the set of possible values. That is: $\{x_{a,k} \mid k \in H\} = \mathcal{V}_a$.

**Clue-Based Constraints:** Each clue in the puzzle introduces additional constraints that must be satisfied by any valid assignment. Note that there are also several implicit positional constraints that must be considered. For example, the leftmost house cannot be on the right of any other house, and the rightmost house cannot be on the left of any other house (as relevant in Clue 4). These spatial constraints, combined with the explicit clues, translate the verbal descriptions into precise logical conditions to be satisfied.. Under the hood, these clues are translated into formal logic formulas that constrain the relationships between variables. For our example puzzle in Figure 2, the constraints can be formulated as follows:

---

### Clue-based Constraints (Example in Figure 2.

**Clue 1.** *"Arnold is not in the first house"*: $x_{\text{Name},1} \neq$ Arnold

**Clue 2.** *"The person who likes milk is Eric"*: $\forall k \in H, (x_{\text{Name},k} = \text{Eric}) \iff (x_{\text{Drink},k} = \text{milk})$

**Clue 3.** *"The photography enthusiast is not in the first house"*: $x_{\text{Hobby},1} \neq$ photography

**Clue 4.** *"The person who loves cooking is directly left of the person who likes milk"*: $\forall k \in H_{<N}, (x_{\text{Hobby},k} = \text{cooking}) \implies (x_{\text{Drink},k+1} = \text{milk})$

**Clue 5.** *"The one who only drinks water is Arnold"*: $\forall k \in H, (x_{\text{Name},k} = \text{Arnold}) \iff (x_{\text{Drink},k} = \text{water})$

**Clue 6.** *"The person who likes milk is not in the second house"*: $x_{\text{Drink},2} \neq$ milk

---

**Task.** The task is to find an assignment of attributes to houses via assigning values to variables $x_{a,k}$ that is consistent with all constraints. These constraints, defined above, include both the uniqueness requirements for attribute values and the logical conditions derived from the specific clues provided. The result is guaranteed to be unique, and can be usually presented as a table as shown in Fig. 2.

## 2.3. ZebraLogic Dataset Creation

To create puzzles, we first define a set of attributes and their corresponding value sets. We also establish some clue types, each with its own language templates containing placeholders for values.

**Attributes and Values.** We construct the attribute set $\mathcal{A}$, which includes the many elements (see Appendix B). Each attribute is associated with a minimum of 6 possible values, ensuring a rich and diverse set of puzzles. Importantly, we always include the `Name` attribute in our samples, as it serves as a crucial element in the puzzle-solving process.

**Clue Types.** The possible clue types are categorized into several types, including FOUNDAT, SAMEHOUSE, NOTAT, DIRECTLEFT/RIGHT, SIDEBYSIDE, LEFT/RIGHTOF, and ONE/TWOBETWEEN. Each clue type captures a specific relationship between variables, providing a diverse set of constraints for the puzzles. More details are in Appendix B.

---

### Clue Types and Illustrative Examples.

- **FOUNDAT**: The tea drinker lives in House 3.
- **SAMEHOUSE**: The musician drinks tea.
- **NOTAT**: The musician does not drink tea (not at the same house).
- **DIRECTLEFT/RIGHT**: The greenhouse is directly to the left/right of the white house.
- **SIDEBYSIDE**: The coffee drinker and the tea drinker are next to each other.
- **LEFT/RIGHTOF**: A is somewhere to the left/right of B.
- **ONE/TWOBETWEEN**: 1/2 houses are between A & B.

---

### Algorithm 1 ZebraLogic Puzzle Generation.

**Require:** A set of possible attributes $\mathcal{A}_{\text{all}}$ and their value sets $\mathcal{V}_a$ for each $a \in \mathcal{A}_{\text{all}}$
**Require:** Clue types $\mathcal{C} = \{c_1, \ldots, c_L\}$ with templates $T(c)$ for each $c \in \mathcal{C}$
**Require:** Number of houses $N$, number of attributes $M$
 1: Sample $M$ attributes from $\mathcal{A}_{\text{all}}$ to form $\mathcal{A} = \{a_1, \ldots, a_M\}$
 2: Initialize solution $S : H \times \mathcal{A} \to \bigcup_{a \in \mathcal{A}} \mathcal{V}_a$ randomly
 3: $C \leftarrow \texttt{ClueGeneration}(S)$       // Initialize clue set
 4: **while** $C \neq \emptyset$ **do**
 5:     $p \leftarrow \texttt{SampleClue}(C)$     // Sample a clue to remove
 6:     $C' \leftarrow C \setminus \{p\}$
 7:     **if** $|\text{Solutions}(C')| = 1$ **then**
 8:         $C \leftarrow C'$      // Remove until $S$ is the unique solution
 9:         **break**
10:     **end if**
11: **end while**
12: **return** $(S, C)$      // Return solution and minimal clue set

---

**Task Generation Algorithm.** Algo. 1 outlines our approach for generating ZebraLogic puzzles. The process starts by sampling $M$ attributes from the full attribute set and creating an initial solution grid $S$ through random value assignments. From this solution, we generate a comprehensive set of clues $\mathcal{C}$ that capture all valid relationships between values in the grid. The algorithm then employs an iterative minimization procedure - at each step, it randomly samples a clue $p \in \mathcal{C}$ and attempts to remove it. Using a SAT solver, it verifies whether the reduced clue set $\mathcal{C}' = \mathcal{C} \setminus \{p\}$ still uniquely determines the original solution $S$. If uniqueness is preserved, $p$ is permanently removed and the process continues. This iteration terminates when no any additional clue can be removed without augmenting the solution space.

We employ weighted sampling during clue selection, assigning higher probabilities to simpler clue types (e.g., FOUNDAT-type clues are more likely to be sampled than NOTAT) to balance puzzle complexity, such that we can efficiently reduce the clue set while maintaining the difficulty of the puzzles. The result is a minimal set of clues that, when combined with the background information about the attributes and their possible values, forms a logically sound puzzle with a single, unique solution. This approach ensures that each generated puzzle is both solvable and challenging, requiring a combination of logical deduction and non-monotonic reasoning strategies to solve. Finally, we use predefined one-shot prompting templates to format the puzzle and instruct the LLMs to generate their reasoning steps and final results in a JSON format (see Appendix D.1).

**Dataset Statistics.** The dataset consists of 1,000 puzzles where the size of the search space varies significantly. The puzzles are based on $N \times M$ grids where $N, M \in \{2, ..., 6\}$ (i.e., 25 sizes in total, with 40 puzzles per size), covering a wide range of complexity. The average and median number of clues per instance is $10.4$ and $9$, respectively.

## 2.4. Theoretical Problem Complexity

By reduction from the Quasigroup (or Latin square) Completion Problem (QCP) (Colbourn, 1984; Gomes & Shmoys, 2002), the ZebraLogic problem is proven to be NP-complete (Sempolinski, 2009). While the problem definition includes a rich set of clue types that can be further expanded, a sufficient condition for the NP-completeness result is to at least include the FOUNDAT and NOTAT clue types. As a result, while a solution to a ZebraLogic puzzle can be easily verified, solving ZebraLogic puzzles for large instances may become intractable within reasonable time frames using current computational methods. This implies that, for a fixed LLM size, the required number of reasoning tokens may increase exponentially with the size of the puzzle.

## 2.5. Measuring Effective Instance Complexity

**Search space size.** We define the solution space of a ZebraLogic puzzle as the total number of possible configurations that can satisfy the uniqueness constraints of the puzzle. That is, a $N \times M$ grid has a solution space of $(N!)^M$, where $N$ is the number of houses and $M$ is the number of attributes. The complexity of the search space increases factorially with the size of the grid, leading to a combinatorial explosion in the number of possible configurations.[2] To better group the puzzles based on their complexity, we categorize them into four groups based on the size of the search space $|\mathcal{S}|$:

- 🟢 **Small** ($|\mathcal{S}| < 10^3$): 2×2, 2×3, 2×4, 2×5, 2×6, 3×2, 3×3, 4×2
- 🟦 **Medium** ($10^3 \leq |\mathcal{S}| < 10^6$): 3×4, 3×5, 3×6, 4×3, 4×4, 5×2, 6×2
- ◆ **Large** ($10^6 \leq |\mathcal{S}| < 10^{10}$): 4×5, 5×3, 4×6, 5×4, 6×3
- ◆◆ **X-Large** ($|\mathcal{S}| \geq 10^{10}$): 5×5, 6×4, 5×6, 6×5, 6×6

**Z3 conflicts.** While search space size provides a useful measure of puzzle scale, it is not the only indicator of complexity. To complement it, we also use the Z3 SMT solver's conflict metric. Z3 (de Moura & Bjørner, 2008) uses the Conflict Driven Clause Learning (CDCL) algorithm, a backtracking approach based on the DPLL (Davis-Putnam-Logemann-Loveland) algorithm. When solving a puzzle, Z3 records the number of conflicts encountered - situations where the solver must backtrack due to contradictions in its current assignment. We run Z3 on each puzzle for 32 times and take the average number of conflicts as a measure of complexity. Puzzles with zero conflicts can typically be solved through simple forward chaining, whereas puzzles with more conflicts require extensive backtracking, indicating higher logical complexity.

While search space size captures the number of candidate assignments (given uniqueness constraints), Z3 conflicts quantify the solver's difficulty in reaching a valid solution.

---

[2]For example, a 3x4 grid has a solution space of $(3!)^4 = 1296$, while a 4x3 grid has a solution space of $(4!)^3 = 13824$.

Together, these metrics offer a complementary view of how the difficulty of the puzzles scales with the problem size. Appendix B provides additional details on how these two metrics vary as a function of the puzzle parameters ($N$, $M$).

## 3. Evaluation

**Setup and Metrics.** Our evaluation is done in a one-shot in-context learning setting, where we provide the models with a single example of how to solve a ZebraLogic puzzle and present the solution in JSON format, and we instruct the LLMs to output their reasoning and solution in the same format, thus making it easier to parse and evaluate their answers. We mainly look at the puzzle-level accuracy, meaning that only when all cells in the grid are filled correctly, the model is considered to have solved the puzzle. In addition to that, we also report the cell-level accuracy.

**Evaluated models.** We evaluate both open-weight LLMs (e.g., Llama and Qwen) and proprietary LLM APIs including GPT-4o, O1 and Claude models. All evaluated models are prompted in the same way (see Appendix D.1), and we use the same greedy decoding and prompts and parsing script across all models to ensure a fair comparison, except for O1, which does not only greedy decoding so we run it three times and take the best result.

### 3.1. Main results

Table 1 shows the performance of various models. o1 outperforms all other models, achieving an overall accuracy of 81.0%, and DeepSeek-R1, an open-weight reasoning LLM achieves 78.7%, with a slightly better performance on Small and Medium-size puzzles than o1-full. However, R1's performance on Large and X-Large puzzles is worse than o1-full. o1-preview and o1-mini achieve 71.4% and 59.7% accuracy, respectively. In contrast, the best-performing open-weight non-reasoning LLM, Sonnet-3.5-1022, only reaches 36.2%. The performance gap is even more pronounced in larger search spaces, where O1-Preview maintains a 17.0% accuracy in the X-Large category, while other models struggle to achieve any correct solutions.

We find that our ranking and scoring of these models are aligned with other reasoning benchmarks such as MATH (Hendrycks et al., 2021) for mathematical reasoning and LiveCodeBench (Jain et al., 2024) for competitive programming. This suggests that the logical reasoning ability of LLMs is highly correlated with their performance on other types of reasoning tasks.

### 3.2. Curse of Complexity in Reasoning with LLMs

We observe that the performance of LLMs drops significantly as the search space size increases, as shown in Fig. 1 and Fig. 3 (in Appendix). We find that for models that are

| Model Names | Overall Grid-level acc. | ● Small $< 10^3$ | ■ Medium $10^3 \sim 10^6$ | ◆ Large $10^6 \sim 10^9$ | ◆◆ X-Large $> 10^9$ | Cell-level Acc. |
|---|---|---|---|---|---|---|
| o1-full 🔒 | 81.0 | 97.2 | 92.1 | 78.0 | 42.5 | 78.7 |
| *DeepSeek-R1* 🔧 | 78.7 | 98.4 | 95.7 | 73.5 | 28.5 | 80.5 |
| o1-preview 🔒 | 71.4 | 98.1 | 88.2 | 59.5 | 17.0 | 75.1 |
| o1-mini 🔒 | 59.7 | 87.5 | 76.8 | 39.0 | 12.0 | 70.3 |
| Claude Sonnet 3.5 🔒 | 36.2 | 84.7 | 28.9 | 4.0 | 1.0 | 54.3 |
| Llama-3.1-405B 🔧 | 32.6 | 81.3 | 22.5 | 1.5 | 0.0 | 45.8 |
| *GPT-4o* 🔒 | 31.7 | 80.0 | 19.6 | 2.5 | 0.5 | 50.3 |
| Gemini-1.5-Pro 🔒 | 30.5 | 75.3 | 20.7 | 3.0 | 0.0 | 50.8 |
| Mistral-Large-2 🔧 | 29.0 | 75.9 | 15.0 | 2.5 | 0.0 | 47.6 |
| Qwen2.5-72B 🔧 | 26.6 | 72.5 | 12.1 | 0.0 | 0.0 | 40.9 |
| Gemini-1.5-Flash 🔒 | 25.0 | 65.0 | 13.6 | 2.0 | 0.0 | 43.6 |
| Llama-3.1-70B 🔧 | 24.9 | 67.8 | 10.4 | 1.5 | 0.0 | 28.0 |
| DeepSeek-v2.5 🔧 | 22.1 | 62.2 | 7.9 | 0.0 | 0.0 | 38.0 |
| *GPT-4o-mini* 🔒 | 20.1 | 58.8 | 4.6 | 0.0 | 0.0 | 41.3 |
| Gemma-2-27B 🔧 | 16.3 | 46.6 | 5.0 | 0.0 | 0.0 | 41.2 |
| Llama-3.1-8B 🔧 | 12.8 | 39.4 | 0.7 | 0.0 | 0.0 | 13.7 |
| Phi-3.5-4B 🔧 | 6.4 | 19.4 | 0.7 | 0.0 | 0.0 | 6.0 |

Table 1: Performance of LLMs on ZebraLogic. The overall accuracy is calculated based on the number of puzzles solved correctly. We also report the accuracy on small, medium, large, and x-large groups based on the size of the search space (see Sec. 2.3). The cell accuracy indicates the percentage of individual cells filled correctly. See Appx. A for more model results.

overall worse than GPT-4o-mini can hardly solve puzzles beyond the Small category — less than 5% accuracy in Medium-size puzzles and almost no correct solutions in Large and X-Large puzzles. We can see that even the largest open-weight LLM, Llama-3.1-405B, only achieves 32.6% overall accuracy. Although 405B has 22.5% accuracy in Medium-size puzzles, it quickly also drops to 1.5% in the Large category and 0.0% in the X-Large category.

The best non-reasoning LLM, Sonnet 3.5, has 36.2% accuracy in the overall evaluation, but it also drops to 4.0% in the Large category and 1.0% in the X-Large category. This indicates that the logical reasoning tasks in ZebraLogic are extremely challenging for LLMs, especially for puzzles with more complexity – with larger search spaces or harder clues. We can also see that scaling up the model size does not necessarily improve the performance of LLMs in logical reasoning tasks with large search spaces.

### 3.3. Scaling Behavior of LLMs in Logical Reasoning

In the following sections, we study the scaling behavior of LLMs in logical reasoning, as illustrated in Fig. 1. Our analysis focuses on two primary types of scaling: 1) scaling model size and 2) scaling test-time compute. For test-time compute, we further explore three sub-dimensions: 1) the number of candidate samples, 2) the number of reasoning tokens (i.e., CoT tokens) generated during inference, and 3)

the sample size for repeated sampling.

## 4. Scaling Model Size Can Hardly Break the Curse of Complexity in Reasoning

**The Curse of Complexity in Reasoning for non-reasoning LLMs.** In addition to the search space size, we also use Z3-conflict as the complexity measure to study the scaling behavior LLMs. Fig. 1 (left) highlights a key observation regarding the performance of various Llama models with different model sizes across an increasing complexity in terms of how many Z3 conflicts on average are encountered when solving the ZebraLogic puzzles. A notable finding is that all model sizes experience a rapid decline in accuracy as the complexity increases, illustrating the challenge posed by complex reasoning tasks. This trend emphasizes the inherent difficulty models face in maintaining high accuracy beyond a certain threshold of search complexity, irrespective of their size. The phenomenon termed as the "curse of complexity" becomes evident as even the largest models, such as the Llama-3.1-405B, cannot sustain high accuracy once the search space surpasses a certain scale. As shown in Fig. 3, we see a similar trend in the search space size.

**Scaling model size is only effective for smaller search spaces.** However, it is important to note the significant benefits of scaling model size when the search space is relatively small (e.g., $\leq 10^6$). In these cases, larger models like the

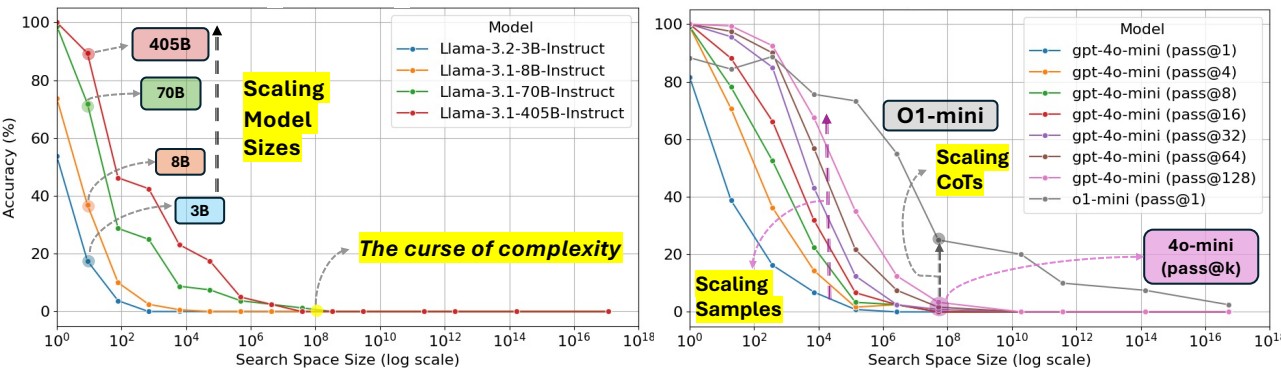

Figure 3: **Accuracy vs Search Space Size** (log scale) comparing multiple scaling behavior of LLMs on ZebraLogic. Left: Scaling model sizes. Right: Scaling test-time compute through two approaches - increasing sample size (via pass@k evaluation) and extending chain-of-thought reasoning length. Both model size and test-time compute show diminishing returns as search space complexity grows beyond a certain complexity. More results are presented in Sec. 3.

Llama-3.1-405B and Llama-3.1-70B demonstrate substantial improvements in accuracy compared to smaller models such as the 3B and 8B versions. This suggests that scaling up the model size is an effective strategy for enhancing performance and tackling reasoning tasks in simpler search spaces. Yet, as the complexity of the search space grows beyond $10^6$, the advantages of larger model sizes diminish, and scaling up the model size proves to be less impactful. This finding underscores the limited utility of model scaling when dealing with highly complex reasoning tasks, as the accuracy plateaus regardless of model size.

**Model Size Scaling Limitations.** This analysis reveals that scaling up model sizes eventually reaches a point of diminishing returns in complex search spaces. Beyond a certain complexity threshold, increasing model parameters is insufficient to prevent performance decline. This highlights a critical boundary for current scaling strategies, suggesting that new approaches are needed to overcome the limitations imposed by high search space complexity and to advance reasoning capabilities further.

## 5. Scaling Test-Time Compute with Repeated Sampling: Promises & Challenges

We examine the impact of scaling test-time compute, a crucial factor affecting LLM performance on logical reasoning tasks. Specifically, here we investigate how increasing the number of candidate samples influences model performance. We begin by employing Best-of-N (BoN) sampling, where we repeatedly sample N candidates from the model for each puzzle. From these candidates, we can select the best answer using various strategies, including majority voting and existing reward models. To understand the theoretical upper bound of this approach, we also analyze BoN sampling with oracle selection, where we use knowledge of the correct

answer to choose the best candidate from the sample pool - equivalent to the pass@k metric in our evaluation (see the right-most plot in Fig. 1 and Fig.3).

**GPT-4o with Best-of-N sampling and oracle selections can achieve nearly o1 performance.** To understand the potential improvement of scaling test-time compute for logical reasoning, we sample 128 candidates from GPT-4o-mini and GPT-4o and study the *coverage* of the correct answer in the sampled candidates. In Table 2, we refer to this coverage metric as *BoN-Oracle*, meaning that the best-of-N (BoN) selection is performed given the oracle knowledge of the correct answer, i.e., the pass@k metric.

We observe that the BoN-Oracle selection can significantly improve the performance of GPT-4o-mini and GPT-4o. For example, GPT-4o with BoN-Oracle$_{N=128}$ achieves an overall accuracy of 69.1%, which is higher than O1-mini's accuracy of 59.7% and a potential scaling effect that can also outperform O1-preview's accuracy of 71.4% if we keep enlarging the sampling size. Note that on the Medium-size examples, we can already see a higher accuracy of 92.9% for BoN-Oracle$_{N=128}$ compared O1-preview's 88.2%, and the trend shown in the curves indicates that the performance of GPT-4o can be further improved with more test-time compute. Fig. 6 in Appendix provides further analysis on how sampling affects model performance.

**Majority Voting is simple yet effective.** For majority voting, we rank the candidates based on the frequency of each cell in their solution grid, and select the candidate with the highest sum of frequencies. As for the Reward Model (RM), we choose the one that ranks to the top on Ai2's RewardBench leaderboard (Lambert et al., 2024b), named Skywork-Reward-Llama-3.1-8B-v0.2 (Liu et al., 2024). We find that using Majority Voting for GPT-4o can improve from 31.7 to 38.0 (for the overall accuracy) when the sam-

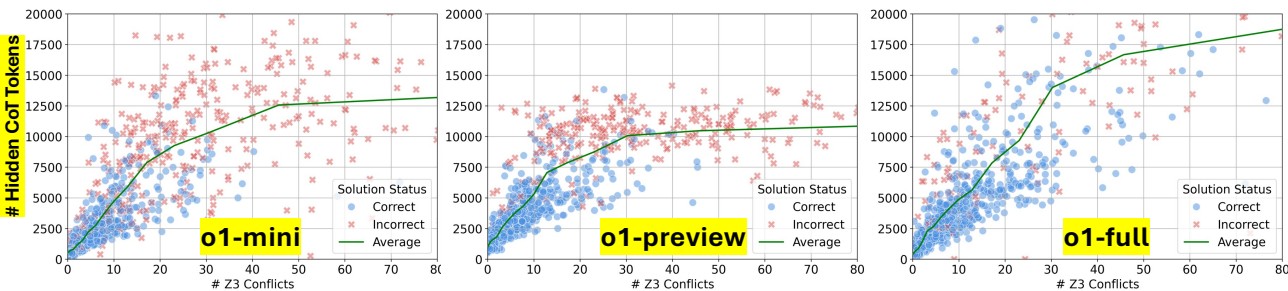

Figure 4: The o1 models' hidden CoT tokens vs. the number of Z3 conflicts. Each point is an example with a certain number of Z3 conflicts. Larger number of Z3 conflicts are associated with harder reasoning problems.

| Model & Methods | | Overall | Small | Medium | Large | X-Large |
|---|---|---|---|---|---|---|
| ◎ **GPT-4o** ↘ | | 31.7 | 80.0 | 19.6 | 2.5 | 0.5 |
| BoN-Oracle$_{N=128}$ 🔮 | | 69.1 | 99.7 | 92.9 | 49.0 | 7.0 |
| BoN-Oracle$_{N=32}$ 🔮 | | 60.3 | 98.4 | 81.1 | 28.0 | 2.5 |
| Majority-Voting$_{N=128}$ | | 37.6 | 84.7 | 32.1 | 7.5 | 0.0 |
| Majority-Voting$_{N=32}$ | | 38.0 | 84.1 | 34.3 | 7.0 | 0.5 |
| BoN-RM$_{N=32}$ | | 33.9 | 77.8 | 28.9 | 4.5 | 0.0 |
| Self-Verify (Oracle) 🔮 | | 34.8 | 83.8 | 24.6 | 5.0 | 0.5 |
| Self-Verify | | 33.0 | 82.2 | 22.1 | 2.5 | 0.0 |
| Self-Verify (x2) | | 32.1 | 80.0 | 21.4 | 2.5 | 0.0 |
| ◎ **GPT-4o-mini** ↘ | | 20.1 | 58.8 | 4.6 | 0.0 | 0.0 |
| BoN-Oracle$_{N=128}$ 🔮 | | 51.2 | 99.7 | 61.8 | 10.0 | 0.0 |
| BoN-Oracle$_{N=32}$ 🔮 | | 42.7 | 97.8 | 39.3 | 2.0 | 0.0 |
| Majority-Voting$_{N=128}$ | | 25.0 | 69.4 | 8.9 | 1.5 | 0.0 |
| Majority-Voting$_{N=32}$ | | 24.5 | 69.1 | 8.2 | 0.5 | 0.0 |
| BoN-RM$_{N=32}$ | | 22.5 | 62.2 | 9.3 | 0.0 | 0.0 |
| Self-Verify (Oracle) 🔮 | | 22.3 | 65.0 | 5.4 | 0.0 | 0.0 |
| Self-Verify | | 21.1 | 60.9 | 5.7 | 0.0 | 0.0 |

Table 2: Comparison of various test-time compute scaling methods applied to GPT-4o and GPT-4o-mini. We evaluate several approaches: BoN-Oracle (selection using oracle knowledge to verify correct answers among samples), BoN-RM (selection using a reward model), Majority-Voting (selecting the most common answer across samples), and Self-Verify (using multi-turn prompting for self-reflection and correction, with and without oracle knowledge). We use 🔮 to denote the use of oracle knowledge.

ple size N=32, while keep increasing the sample size does not necessarily improve the performance any more. Also, the performance of GPT-4o with BoN-RM$_{N=32}$ is 33.9, which is worse than majority voting, suggesting that the current reward models that are mainly designed for chat or general instruction following tasks may not be directly applicable to (logical) reasoning tasks.

## 6. Scaling Test-Time Compute with Extensive Chain-of-Thoughts Tokens

Another approach of scaling test-time compute is to increase the number of reasoning tokens (i.e., chain-of-thoughts tokens) that the model generates during inference.

### 6.1. o1 Generates More Hidden Reasoning Tokens

**o1 generates large-scale hidden reasoning tokens.** One of the key differences between o1 and other LLMs is the way they use more test-time compute to decode much more hidden chain-of-thoughts (CoT) tokens during inference time, which are not directly visible to users. Our analysis shows that o1 models scale their hidden CoT tokens with puzzle complexity - producing on average 5,144.6 (o1-mini) and 5,346.3 (o1-preview) hidden reasoning tokens compared to 502.9 and 543.7 for GPT-4o-mini and GPT-4o respectively. This order of magnitude difference in reasoning steps appears to contribute to o1's superior performance on logical reasoning tasks. For detailed analysis of how hidden CoT tokens vary with puzzle complexity, see Appendix C.3.

Figure 4 reveals a positive correlation between the number of hidden reasoning tokens generated by o1-preview and Z3 conflicts, aligning with our earlier observation that o1 allocates more reasoning tokens to more complex puzzles. For puzzles with fewer than 20 Z3 conflicts, we observe a consistent ratio of approximately 400 hidden reasoning tokens per conflict. However, this scaling pattern plateaus when Z3 conflicts exceed 30, suggesting that o1-preview may have reached its maximum reasoning capacity at the current model size. This suggests that while o1-preview can effectively leverage more reasoning tokens for complex puzzles, there is a limit to the extent to which it can scale reasoning tokens to address highly complex reasoning tasks. With the recent release of o1-full, we find that our previous estimation is consistent with the actual number of hidden reasoning tokens generated by o1-full, which is around 5,000 on average. This further confirms the scaling behavior of o1 models in generating more hidden reasoning tokens for complex puzzles.

We also find that when o1-preview make mistakes, they usually generate more hidden reasoning tokens than when they solve the puzzles correctly, which is consistent with the observation that o1 tends to generate more reasoning tokens for more complex puzzles that are harder to solve.

## 6.2. Self-Refinement is Limited but Promising

The other feature of o1's hidden reasoning process is the ability to reflect on its own reasoning process and refine its answer. From our observation on the summary of their hidden reasoning process, we can see that o1 often revisits the clues and constraints to verify its previous reasoning and fix the errors if there are any, which is similar to the Z3 solver's conflict-driven clause learning mechanism. In order to elicit such self-refinement behavior from LLMs, we add follow-up queries to ask the model to review its initial answer and check the clues and constraints again in a multi-turn conversation setting. There are two settings for the self-refinement process: one with the oracle knowledge of the correct answer and the other without the oracle knowledge. Results in Table 2 show modest improvements with self-verification, particularly without oracle knowledge (4o improves from 31.7 to 33.0, then decreases to 32.1).

---

**Self-Verification Prompt**

**Self-Verify:** *Your answer may be incorrect! Identify any mistakes in your reasoning and answer, if any. Correct them to ensure they align with the given information. Present your updated response in the same JSON format mentioned in the initial prompt.*

**Self-Verify (Oracle 💡):**
- **For *incorrect* results:** *Your answer is incorrect! Re-examine the clues, correct the mistakes, and then provide the revised solution in the original JSON format.*
- **For *correct* results:** *Your answer is correct. Please repeat the json-formatted output again.*

---

## 7. Related Work

**Logical Reasoning Benchmarks and Dataset Creation**
Logical reasoning has long been a critical area of AI, but only recently have LLMs been subjected to rigorous testing in this domain. LogiQA (Liu et al., 2020) emerged early on to evaluate complex logical comprehension in question-answering formats; and subsequent efforts by (Liu et al., 2023) reframed it as a Natural Language Inference (NLI) task to further stress-test LLMs' capabilities. Researchers have also explored generating more dynamic or granular datasets to push the limits of reasoning systems. For instance, Madusanka et al. (2024) investigated satisfiability tasks formulated in natural language, studying how varying computational complexity influences LLM inference performance. Similarly, Richardson & Sabharwal (2022) introduced a systematic methodology for building challenging reasoning datasets, exposing robustness gaps in transformer-based models when tasked with increased complexity. Prior work on logic grid puzzles include Mitra & Baral (2015) that proposed a grid-based puzzle dataset prior to the LLM era and focused on automatic translation from language to

a formal specification, Dziri et al. (2023) that investigated compositionality in LLMs on grid-based puzzles, as well as Tyagi et al. (2024) that provided a new error taxonomy to evaluate the correctness of the reasoning chains of LLMs.

**Approaches to Logical Reasoning in LLMs.** Several lines of research propose methods to augment or refine LLMs for stronger logical reasoning. Clark et al. (2020) demonstrated that transformers can emulate logical reasoning over natural language sentences—serving as "soft theorem provers." Pan et al. (2024) showed that a decoder-only Transformer could tackle SAT problems, paralleling the Davis–Putnam–Logemann–Loveland (DPLL) algorithm, thereby expanding the role of LLMs to more complex problem-solving domains. Alternatively, neuro-symbolic systems like CLOVER (Ryu et al., 2024) integrate LLMs with symbolic solvers to better capture the translation of intricate logical semantics from text.

**Empirical Evidence of LLM Limitations**. Despite these promising developments, LLMs face persistent hurdles as logical problem complexity increases. Yan et al. (2024) contended that models may rely heavily on probabilistic correlations rather than genuinely understanding logical rules. Similarly, Xie et al. (2024) highlighted the complex interplay between training data memorization and genuine reasoning abilities of LLMs. Additionally, Schlegel et al. (2022) conducted an extensive empirical study to investigate the detection of formally valid inferences in controlled fragments of natural language, revealing that transformers often overfit to superficial patterns rather than acquiring logical principles. Lam et al. (2024) showed the impact of the choice of symbolic solvers on the effectiveness of LLMs in deductive reasoning tasks, calling for more consistent comparative studies. Further empirical evidence from Dziri et al. (2023) and Parmar et al. (2024) demonstrated that even ostensibly simple logical tasks continue to challenge these models. Finally, Madusanka et al. (2023) investigated the limits of transformers on solving the problem of model-checking with natural language and the significant impact of the language fragment on the performance of transformers.

## 8. Conclusion

We introduce *ZebraLogic*, a controlled benchmark of logic grid puzzles that highlights the scaling limits of LLM-based reasoning through carefully adjustable complexity. Our experiments reveal a pronounced drop in performance as complexity increases, overshadowing gains from model growth or training data expansions. While increasing the generation sample size yields modest improvements, a backtracking-based approach with expanded reasoning steps significantly boosts accuracy. These results spotlight the importance of non-monotonic reasoning and provide a valuable framework for advancing logical reasoning research.

## Impact Statement

Our work introduces a novel, systematically controlled evaluation framework for LLMs based on logic grid puzzles. This framework enables rigorous assessment of logical reasoning capabilities, independent of domain knowledge or data leakage.

A key finding is the "curse of complexity," where LLM accuracy significantly declines as problem complexity increases. This limitation persists despite scaling model size, challenging the notion that larger models inherently improve complex logical reasoning. It suggests inherent architectural constraints in handling non-monotonic reasoning and extensive backtracking.

We also explore test-time compute scaling. While some sampling strategies show diminishing returns, the observation that models like o1 utilize significantly more hidden chain-of-thought (CoT) tokens, scaling with problem complexity, is crucial. This highlights the importance of explicit, structured reasoning processes with backtracking. It redirects future research towards developing LLMs specifically trained for deliberate, comprehensive reasoning steps, potentially via reinforcement learning, rather than solely relying on increased model size.

ZebraLogic serves as a valuable benchmark, offering a quantifiable environment to push LLM reasoning boundaries. Our results provide a critical roadmap for advancing AI, guiding the development of more capable, reliable, and interpretable LLMs for complex logical challenges.

## Acknowledgments

Yejin Choi's research is supported in part by the National Science Foundation under Grant DMS-2134012.

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

## A. Additional Experimental Results and Analysis

Please find the additional analysis and results below in the figures.

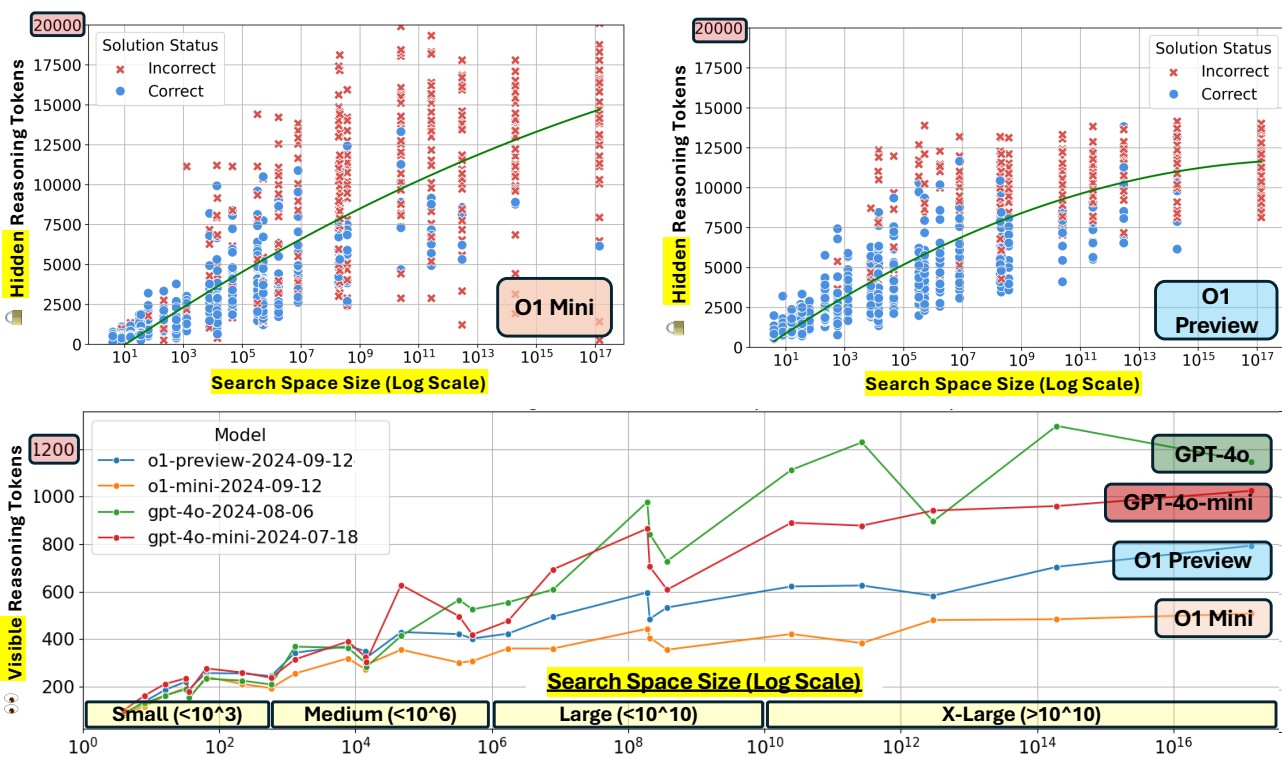

Figure 5: Top: Distribution of hidden reasoning tokens generated by o1-mini and o1-preview models. Bottom: Distribution of visible reasoning tokens across GPT-4o-mini, GPT-4o, o1-mini, and o1-preview models. Mean hidden reasoning tokens per model: o1-mini generates 5,144.6 tokens and o1-preview generates 5,346.3 tokens. Mean visible reasoning tokens per model: GPT-4o-mini (502.9), GPT-4o (543.7), o1-mini (305.7), and o1-preview (402.4).

## B. Details of the ZebraLogic Dataset

All possible attribute types: *Name, Color, Nationality, Animal, Drink, Cigar, Food, Flower, PhoneModel, Children, Smoothie, Birthday, Occupation, Height, CarModel, FavoriteSport, MusicGenre, BookGenre, HairColor, Mother, HouseStyle, Education, Hobby, Vacation, Pet*

Each problem instance is characterized by two complimentary complexity metrics: the search space size as well as the average number of Z3 conflicts that the SMT solver takes to solve a puzzle. Figure 7 illustrates how both metrics vary across different number of houses ($N$) and number of attributes ($M$).

## C. Additional Analysis

**GPT-4o tends to generate more *visible* reasoning tokens than o1.** Interestingly, we find that the GPT4o model tends to generate more visible reasoning tokens than o1, especially when the search space is large, which is shown in the lower part of Figure 5. The visible reasoning tokens are generated by the model and displayed in their outputs before the final solution grids. We can see that until the search space reaches the Large category (especially when the search space size is $< 10^5$), the four models generate similar numbers of visible reasoning tokens. However, when the search space size is larger, GPT4o generates more visible reasoning tokens yet still fails to solve the puzzles. o1 models, which have used more hidden CoT tokens, tend to output fewer visible reasoning tokens for describing their reasoning process.

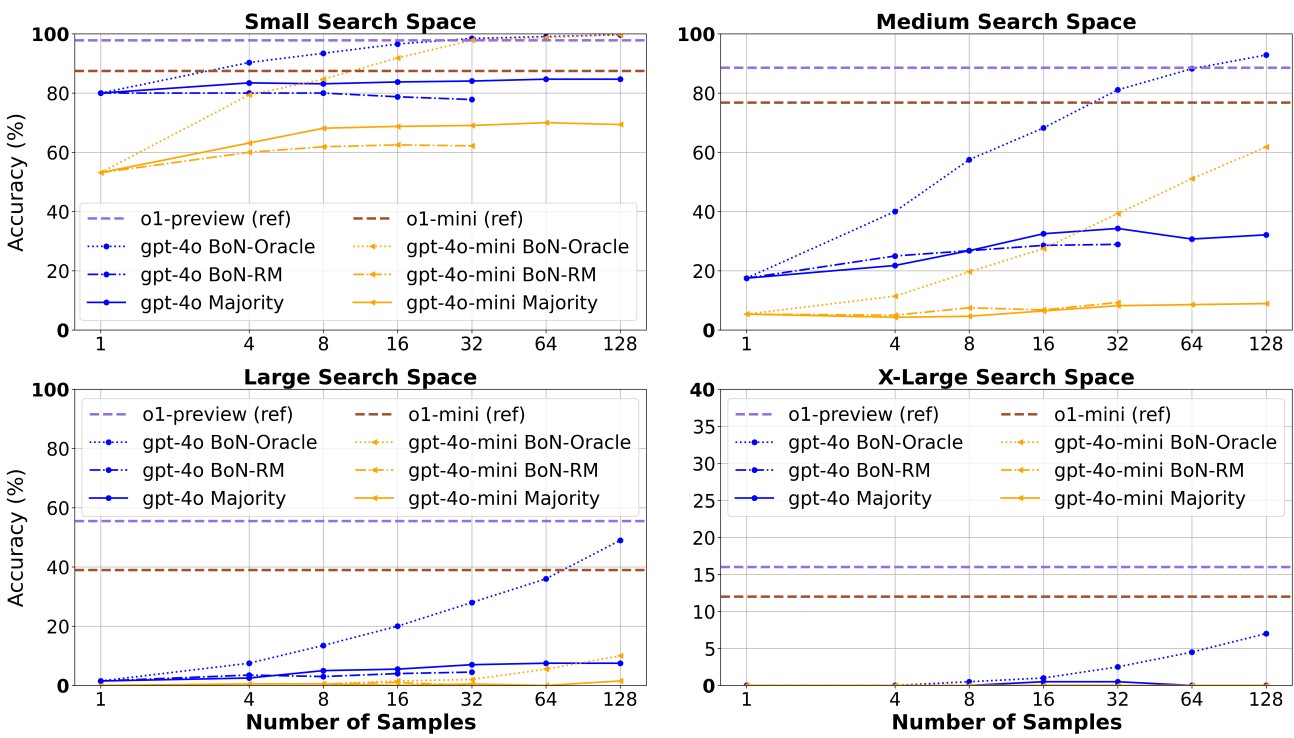

Figure 6: Analysis of inference-time compute scaling using Best-of-N (BoN) sampling across different ZebraLogic puzzle size groups. The curves demonstrate how increasing the number of samples affects model performance, with separate plots for Small, Medium, Large, and X-Large puzzle categories.

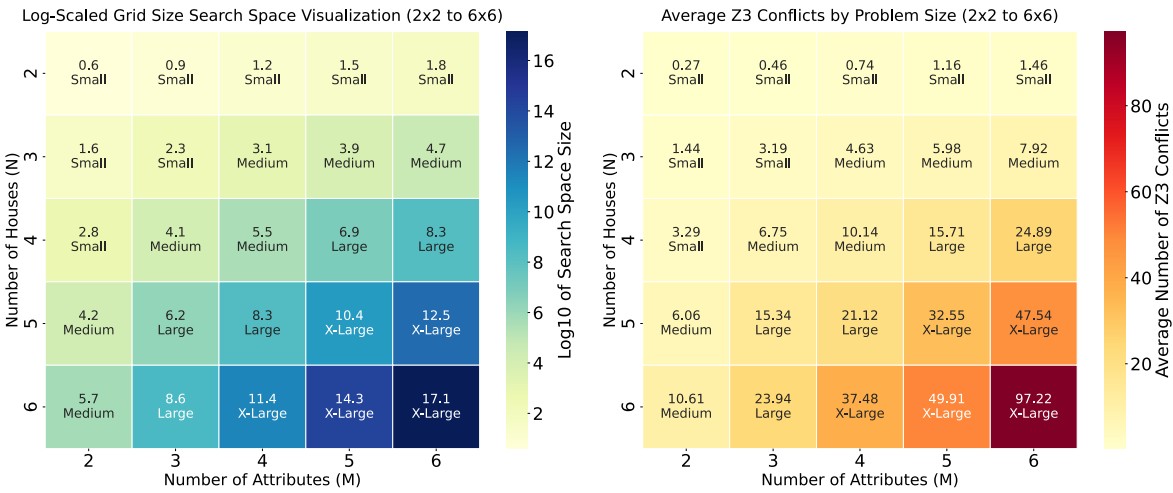

Figure 7: Heatmaps illustrating puzzle complexity metrics across different ZebraLogic problem sizes. The left heatmap represents the log-scaled search space size, categorized from Small to X-Large based on the grid configurations (houses × attributes). The right heatmap shows the average number of Z3 conflicts encountered during solving, with higher counts indicating greater logical complexity.

## C.1. Human Evaluation of o1's Reasoning

Here we present several case studies to understand the reasoning process of o1. We selected a few representative examples from the ZebraLogic dataset and analyzed the reasoning steps taken by o1-preview to arrive at the final solution.

## C.2. Comparison with LMSYS Arena Rankings.

While the overall performance rankings on ZebraLogic generally align with those from the LMSYS Arena (a platform for evaluating LLMs across various tasks), we observe some notable discrepancies that highlight ZebraLogic's distinct evaluation perspective. For instance, GPT-4o-mini-0718 achieves a higher Elo score (1273) in LMSYS Arena (24-11-11) compared to Llama-3.1-405B (1266), GPT-4o-0806(1264), Mistral-Large-2 (1251), and Llama-3.1-70B (1247). However, on ZebraLogic, GPT-4o-mini only achieves 20.1% accuracy while Llama-3.1-405B reaches 32.6%. These differences suggest that ZebraLogic offers a more focused assessment of logical reasoning capabilities, providing valuable insights that complement general-purpose evaluations.

## C.3. o1 generates large-scale hidden reasoning tokens.

One of the key differences between o1 and other LLMs is the way they use more test-time compute to decode much more hidden chain-of-thoughts (CoT) tokens during inference time, which are not directly visible to users. Figure 5 shows how the number of hidden CoT tokens varies with the search space size for both o1-mini and o1-preview. In each sub-figure on the top, we plot 1,000 points, each representing a puzzle. The color and shape of the points indicate whether the model produced a correct solution (blue dots) or an incorrect one (red crosses). The y-axis shows the number of hidden CoT tokens generated by the model, while the x-axis shows the search space size in logarithmic scale. The definition of search space size is provided in Section 2.3, and a larger search space usually indicates a more complex puzzle.

We can see that the number of hidden CoT tokens generated by o1 is scaling with the search space size, indicating that o1 is able to leverage more reasoning steps when faced with more complex puzzles. On average, we find that o1-mini generates 5,144.6 hidden reasoning tokens, while o1-preview generates 5,346.3 hidden reasoning tokens. Both are about 10 times more than the average number of reasoning tokens generated by GPT-4o-mini (502.9) and GPT-4o (543.7), showing that scaling reasoning tokens can be an effective way to improve the performance of LLMs on logical reasoning tasks.

# D. Further Discussion on o1's Reasoning

We have seen that o1 generates more hidden reasoning tokens than other LLMs, and the hidden reasoning tokens scale up with search space size, indicating that o1 is able to leverage more reasoning steps when faced with more complex puzzles. Since the hidden reasoning tokens are not accessible, we investigate whether o1's visible output tokens or its summary of hidden tokens can explain its higher performance.

**Visible outputs from o1 cannot fully explain its reasoning for complex problems.** To understand how o1 reasons, we have to focus on their public reasoning steps that we can extract from the model's visible outputs. From our human evaluation on their reasoning steps, we find that o1's reasoning steps are not necessarily rigorous or complete, even when they arrive at the correct solution. For small-to-medium search spaces, o1-preview's reasoning chains tend to be complete, while o1-mini sometimes can skip some steps to directly reach the solution. For problems with larger search spaces, o1's visible reasoning chains tend to be very incomplete, and sometimes even incorrect, especially when the reasoning process requires backtracking. For example, o1's visible reasoning may contain steps such as "Bob cannot be in Houses 1, 4, or 5, so he must be in House 3" without explaining why Bob cannot be in House 2, although it will indeed lead to the correct solution. Note that such cases also happen for other LLMs such as GPT-4o. We thus describe that the reasoning process of LLMs and o1 models are sometimes based on guessing without formal logic, especially for complex problems with large search spaces, rather than rigorous logical reasoning.

Such incomplete reasoning steps are very common in o1's outputs, especially for puzzles with larger search spaces, leading to unreliable explanations of their reasoning process. Thus, we argue that the visible reasoning steps from o1 cannot help us understand how o1 reasons for complex problems. Furthermore, knowledge distillation from o1's reasoning steps is not necessarily helpful for improving the performance of other LLMs, as the reasoning steps are often incomplete and sometimes incorrect. This raises questions about the concern of hidden CoT tokens in their reasoning process that are not visible in the output.

**Will the summary of hidden tokens help us understand o1's reasoning?** Although the hidden CoT tokens are not visible from the OpenAI APIs, we can see an overview summary of the hidden reasoning tokens on ChatGPT's user interface for o1's hidden reasoning steps. By manually analyzing the overview summary of hidden reasoning tokens, we find it is still hard to clearly understand how o1 reasons for complex problems. We can sometimes see some intermediate results in the overview but not any explanations for the decision. Interestingly, we can see some behaviors of recognizing the

contradictions of previous assumptions and revisiting the clues to refine the solution. Such an in-context reflection behavior is hardly noticeable in other LLMs such as GPT-4o's reasoning, and it may be a key factor for o1's success in solving complex problems. Typical steps in o1's hidden reasoning include: "Laying out the options", "Piecing together clues", "Pinpointing the clues", "Reevaluating assumptions", "Revisiting clues.", "Mapping out connections", "Tracking movement", etc. We provide case studies in the Appendix to better understand how o1 reasons.

## D.1. Prompt template to evaluate ZebraLogic

```
# Example Puzzle

There are 3 houses, numbered 1 to 3 from left to right, as seen from across the street.
↪   Each house is occupied by a different person. Each house has a unique attribute for
↪   each of the following characteristics:
    - Each person has a unique name: `Peter`, `Eric`, `Arnold`.
    - Each person has a unique favorite drink: `tea`, `water`, `milk`

## Clues for the Example Puzzle

1. Peter is in the second house.
2. Arnold is directly left of the one who only drinks water.
3. The one who only drinks water is directly left of the person who likes milk.

## Answer to the Example Puzzle

{
    "reasoning": "Given Clue 1, we know Peter is in House 2. According to Clue 2, Arnold
        is directly left of the one who only drinks water. The person in House 3 cannot
    ↪   be on the left of anyone, so Arnold must be in House 1. Thus, Peter drinks water,
    ↪   and Eric lives in House 3. Then, according to Clue 3, Eric drinks milk.
    ↪   Therefore, Arnold drinks tea.",
    "solution": {
        "House 1": {
            "Name": "Arnold",
            "Drink": "tea"
        },
        "House 2": {
            "Name": "Peter",
            "Drink": "water"
        },
        "House 3": {
            "Name": "Eric",
            "Drink": "milk"
        }
    }
}

# Puzzle to Solve

There are 3 houses, numbered 1 to 3 from left to right, as seen from across the street.
↪   Each house is occupied by a different person. Each house has a unique attribute for
↪   each of the following characteristics:
    - Each person has a unique name: `Eric`, `Peter`, `Arnold`
    - Each person has a unique favorite drink: `milk`, `water`, `tea`
    - Each person has a unique hobby: `photography`, `cooking`, `gardening`

## Clues:
1. Arnold is not in the first house.
2. The person who likes milk is Eric.
3. The photography enthusiast is not in the first house.
4. The person who loves cooking is directly left of the person who likes milk.
5. The one who only drinks water is Arnold.
6. The person who likes milk is not in the second house.

# Instruction

Now please solve the above puzzle. Present your reasoning and solution in the following
↪   json format:

{
    "reasoning": "___",
    "solution": {
        "House 1": {
            "Name": "___",
            "Drink": "___",
            "Hobby": "___"
        },
        "House 2": {
            "Name": "___",
```

```
            "Drink": "___",
            "Hobby": "___"
        },
        "House 3": {
            "Name": "___",
            "Drink": "___",
            "Hobby": "___"
        }
    }
}
```

