# OpenReview forum: "ZebraLogic: On the Scaling Limits of LLMs for Logical Reasoning"
_ICML.cc/2025/Conference — ICML 2025 poster_

### Official Review · Reviewer_LkaF · 2025-03-10

**Overall Recommendation:** 2

**Summary:**

The paper proposes ZebraLogic, a newly developed benchmark dataset of logic grid puzzles derived from constraint satisfaction problems.  The authors systematically evaluate LLM performance across different levels of problem complexity. The authors show "curse of complexity", where model accuracy declines significantly as puzzle complexity increases. The study also explores some strategies to improve reasoning performance, including Best-of-N sampling. The results suggest that scaling model size and test-time compute is insufficient to achieve reasoning in current LLMs.

**Claims And Evidence:**

Why Self-Refinement is promising? It seems that majority voting (or self consistency) has better results than self-refinement.

**Essential References Not Discussed:**

N/A

**Experimental Designs Or Analyses:**

Yes.

**Methods And Evaluation Criteria:**

The paper doesn't provide new methods.

**Other Comments Or Suggestions:**

N/A

**Other Strengths And Weaknesses:**

S1. The paper systematically evaluates various LLMs through extensive experiments.

S2. The proposed dataset is engaging and valuable.

S3. The paper is well-structured and easy to follow.

W1. Introducing a novel method inspired by the experimental results would strengthen the paper.

W2. The Best-of-N sampling with oracle selection has limited practical value. In most reasoning tasks, obtaining an oracle verifier is challenging.

W3. Reasoning capabilities in LLMs encompass various aspects, but the paper primarily focuses on logical reasoning. It would be beneficial to discuss other perspectives.

W4. Many observations in the paper align with well-known trends. For example, the significant performance drop as task complexity increases is expected. Providing deeper insights beyond these common trends would enhance the contribution.

**Questions For Authors:**

N/A

**Relation To Broader Scientific Literature:**

The paper proposes a new dataset for evaluating LLM reasoning capabilities.

**Theoretical Claims:**

There is no theoretical claim.

---

> ### Author Rebuttal · Authors · 2025-04-01
>
> Thank you for your review! We value your constructive feedback and will address your suggestions in the revised version.
>
>
> ---
> #### Q1: Why Self-Refinement is promising? ...
>
> Both self-refinement and majority voting are methods aimed at enhancing the reasoning performance of LLMs, and they are orthogonal to each other. For instance, self-refinement can be used to generate multiple outputs, which can then be evaluated through majority voting to select the most likely output. However, our goal is not to develop a state-of-the-art model for solving these puzzles but to demonstrate that all commonly used approaches, including self-refinement and majority voting, struggle to some extent with these challenges.
>
> The reason we explore self-refinement is that reasoning models like O1 and R1 indicate that LLMs exhibit stronger reasoning capabilities when allowed to reflect on their previous steps, identify errors, and self-correct. Our focus is on understanding the scaling behavior of self-refinement (with multi-turn prompting) and how it compares to majority voting, highlighting the limitations shared across these approaches.
>
> ---
>
> #### W1
>
> We believe our insights will encourage the development of better logical reasoning models, and we will discuss their implications for future research in the revised version.
>
> Here are key points and experiments we will add:
>
> - **Improved Training:** We will generate numerous training examples and explore reinforcement learning methods like GRPO to boost model reasoning. We will also test if training on ZebraLogic puzzles generalizes to domains like math and code generation.
>
> - **Improved Inference:** We will investigate inference techniques to enhance reasoning, such as forced re-verification prompting and refined best-of-N sampling strategies.
>
> ---
>
> #### W2
> We agree that Best-of-N sampling with oracle selection has limited practical value and do not propose it as a solution for reasoning improvement. Instead, we aim to highlight the difficulty of these reasoning tasks by testing the limits of repeated sampling, even with a perfect oracle. This shows that current models struggle under ideal conditions, underscoring the tasks' challenge. We respectfully disagree that this is a paper weakness but will clarify this point to avoid confusion.
>
> ---
>
> #### W3
>
> We recognize that LLM reasoning includes domains like spatial, causal, and analogical reasoning beyond logical reasoning. However, we argue that logical reasoning is a foundational aspect of intelligence, justifying it as a critical starting point. Its structured nature enables precise evaluation and controlled experiments, vital for broader reasoning insights.
>
> Additionally, our methodology for creating controllable reasoning problems is adaptable to other domains. For example, in spatial reasoning, we could generate grids with constraints for models to infer relationships. Many reasoning tasks share a logical basis involving constraint-solving, suggesting our approach provides a versatile framework for advancing various LLM reasoning types.
>
> ---
>
> #### W4
>
> We appreciate the feedback and agree that performance drops with task complexity across domains. However, we contend that the extent and nature of this decline in logical reasoning for LLMs are underexplored, which our work systematically examines.
>
> Our paper provides new insights into LLM scaling limits in logical reasoning, including:
>
> 1. **Quantifiable Complexity Metrics:** We introduce ZebraLogic, a 1,000-puzzle benchmark, using search space size and Z3 conflict count to explain performance drops, offering insights beyond prior studies (Sec. 2.3, Fig. 8).
>
> 2. **Scaling Behavior Analysis:** We examine model size, sampling, and test-time compute, showing that even advanced methods (e.g., Llama-3.1-405B, pass@128) fail past certain complexity levels, questioning the "more scale" assumption (Sec. 4-6, Fig. 1).
>
> 3. **Reasoning Token Insights:** Our analysis of OpenAI’s o1 reveals heavy use of hidden chain-of-thought tokens (up to 10x more than GPT-4o), yet performance plateaus at high complexities, indicating inference-time reasoning trade-offs (Sec. 6, Fig. 3).
>
> 4. **Practical Implications:** We evaluate strategies like Best-of-N sampling and self-verification, noting their limits and proposing explicit step-by-step reasoning to enhance LLM capabilities (Sec. 5-6).
>
>
> In summary, while the performance drop with complexity may align with general expectations, our work offers a rigorous, systematic dissection of this trend in logical reasoning, uncovering its boundaries and underlying causes. These contributions provide deeper insights that extend well beyond common observations, positioning ZebraLogic as a valuable tool for both understanding and addressing the reasoning limitations of LLMs.
>
> ---
>
> Thank you again for the review! We will address your suggestions in the revised version. Please contact us with any further questions.

---

### Official Review · Reviewer_gf4p · 2025-03-14

**Overall Recommendation:** 3

**Summary:**

The paper introduces ​ZebraLogic, a benchmark dataset of 1,000 logic grid puzzles derived from constraint satisfaction problems (CSPs), to evaluate the scalability of large language models (LLMs) in complex non-monotonic reasoning. Key findings include the curse of complexity, and scaling model size or test-time compute (e.g., sampling, backtracking) offering limited improvements. The study evaluates GPT-4o, Llama-3, and specialized reasoning models (o1, R1).

**Claims And Evidence:**

Mostly yes.

**Essential References Not Discussed:**

N/A

**Experimental Designs Or Analyses:**

Yes.
The main experiment and test-time scaling experiment are reasonable.
However, the paper lacks error analysis, such as qualitative failure case studies.

**Methods And Evaluation Criteria:**

Yes. The evaluation criteria are reasonable, including both puzzle and cell-level accuracy.

**Other Comments Or Suggestions:**

1. Figure 3 needs more captions to describe the results briefly.

**Other Strengths And Weaknesses:**

Strengths:
1. The paper isolates logical reasoning from domain knowledge, ensuring a controlled evaluation.
2. The paper uses two complementary metrics to define problem complexity: search space size and Z3 conflicts.
3. The experiments test on a broad range of models. The curse of complexity is effectively demonstrated.
4. Various inference-time strategies are tested, including Best-of-N sampling, majority voting, and self-verification. Results find that scaling alone is insufficient.

Weaknesses:
1. The paper does not deeply analyze why scaling fails.
2. The paper lacks a qualitative analysis of model errors.
3. The paper assumes that Z3 conflicts correlate with reasoning difficulty but does not experimentally validate this claim.
4. The paper only evaluates models in a one-shot setting, which might not be optimal for logical reasoning.

**Questions For Authors:**

1. Can you briefly discuss why scaling fails?
2. Does the number of hidden reasoning tokens generated by o1 models correlate with puzzle difficulty?
3. Why does Best-of-N sampling with oracle selection significantly improve performance, while reward models fail to be as effective?

**Relation To Broader Scientific Literature:**

The paper is related to LogiQA (QA-style logical reasoning) and CLOVER (Neuro-symbolic LLM-solver integration).

**Theoretical Claims:**

N/A

---

> ### Author Rebuttal · Authors · 2025-04-01
>
> Thank you very much for your thoughtful review! We address each of the concerns and questions raised, and we will incorporate these clarifications and additional analyses into the camera-ready version of the paper. We hope our responses adequately resolve the issues highlighted and kindly ask for your reconsideration of the scores if appropriate.
>
>
> #### W1: The paper does not deeply analyze why scaling fails
> Due to page constraints, our initial submission focused on empirical observations, but we plan to enhance the revised paper with a detailed examination of failure modes. This will include: (1) analysis of the number and types of unsatisfied constraints in model outputs, and (2) investigation of the correlation between clue ordering and solution correctness. These additions will provide deeper insights into the limitations of scaling for logical reasoning in LLMs. We also did qualitative analysis and discussed them in the response to W2.
>
>
> #### W2: The paper lacks a qualitative analysis of model errors
>
> Thanks for the suggestion! While we included some failure case studies (e.g., Appendix C.1), page limits constrained a full qualitative analysis in the main text. We will address this in the revised version by integrating key error patterns:
>
> - **Non-Reasoning Models:** Frequent hallucinations in initial and later steps (e.g., Llama-3.1-405B) lead to inconsistent deductions and incorrect outputs.
> - **Reasoning Models:** Fewer early hallucinations (e.g., o1, R1), but errors persist in later steps due to incomplete backtracking.
> - **Self-Verification:** Reasoning models excel with self-correction (e.g., R1’s “Wait”/“Correct” markers, o1’s clue revisits), absent in non-reasoning models.
> - **Clue Rephrasing:** Periodic rephrasing of clues by reasoning models enhances constraint understanding and reduces errors.
>
>
>
> #### W3: The paper assumes that Z3 conflicts correlate with reasoning difficulty but does not experimentally validate this claim
>
> Thanks for the question. We recognize that "reasoning difficulty" has no universal definition and appreciate concerns about Z3 conflicts as a general indicator beyond Z3. To clarify, Z3 conflicts reflect the "reasoning difficulty of Z3," a leading systematic heuristic solver, though not fully solver-agnostic like search space size. Still, we find it insightful. Search space size, like grid size in our study, indicates challenges for uninformed reasoners, where a 10x increase means 10x more brute-force effort. Likewise, Z3 conflicts gauge difficulty for advanced solvers like Z3. Although we didn't quantitatively validate Z3 conflicts' correlation with perceived reasoning difficulty, our results using Z3 conflicts and grid size as proxies match scaling trends in benchmarks like AIME regarding model size and inference cost.
>
>
> #### W4: The paper only evaluates models in a one-shot setting, which might not be optimal for logical reasoning
>
> In our preliminary experiments, we find that providing more few-shot examples do not improve the performance. We will include the results in the revised paper. It is indeed more likely to improve the performance for smaller non-reasoning models. However, our focus in the paper is to study the scaling behavior of reasoning models, instead of finding the best prompting strategies or exact few-shot examples to get state-of-the-art performance on reasoning benchmarks.
>
> #### W5: The supplementary material lacks environment requirements and a readme file, making it hard to reproduce
> Thanks for pointing out! We will add a README file to the repository to help the readers to reproduce the results. We believe the code is not too complicated to run. The only complex part is to parse model outputs and calculate the metrics, while the other parts are standard LLM inference and data processing.
>
> #### Q1: Can you briefly discuss why scaling fails?
> Please refer to the answer for W1.
>
> #### Q2: Does the number of hidden reasoning tokens generated by o1 models correlate with puzzle difficulty?
>
> Yes, as shown in Figure 6 and discussed in Section 6.1, the number of hidden reasoning tokens generated by o1 models is positively correlated with puzzle difficulty, where difficulty is reflected by the Z3 conflict metric as an indicator of the reasoning challenges faced by Z3.
>
> #### Q3: Why does Best-of-N sampling with oracle selection significantly improve performance, while reward models fail to be as effective?
>
> Oracle selection picks the model output closest to the ground truth, boosting performance as it mimics using ground truth as a reward signal, which is impractical. We study reward models to explore their limits in evaluating LLM reasoning performance.
>
> ---
>
> Thank you for your review! We will address your suggestions in the revised version. We hope these clarifications and additional analyses adequately resolve the issues highlighted and kindly ask for your reconsideration of the scores if appropriate.

---

### Official Review · Reviewer_KS5B · 2025-03-14

**Overall Recommendation:** 4

**Summary:**

The paper introduces ZebraLogic, a benchmark of logic grid puzzles derived from constraint satisfaction problems (CSPs), to evaluate the logical reasoning capabilities of LLMs. Key findings include:

1. Curse of Complexity: LLM performance declines sharply as puzzle complexity (measured by search space size and Z3 solver conflicts) increases, even with model scaling or test-time compute.

2. Scaling Limitations: Larger models (e.g., Llama-3.1-405B) improve performance on simpler puzzles but fail in highly complex scenarios, suggesting fundamental reasoning limitations.

3. Hidden Reasoning Tokens: Models like o1 generate significantly more hidden chain-of-thought (CoT) tokens during inference, correlating with improved performance, though scaling plateaus at high complexity.

4. Test-Time Compute: Best-of-N sampling and self-verification prompts yield marginal gains but fail to overcome the curse of complexity.

**Claims And Evidence:**

Yes

**Essential References Not Discussed:**

None

**Experimental Designs Or Analyses:**

- Broad evaluation of open/closed-source models across complexity levels.
- Lack latest models like o3-mini.

**Methods And Evaluation Criteria:**

- ZebraLogic is well-designed, using CSPs to isolate logical reasoning from domain knowledge.
- Complexity metrics (search space size, Z3 conflicts) are appropriate.
- Prompting / Best-of-N sampling / Voting / Self-Verify metrics are common practice.

**Other Comments Or Suggestions:**

1. Typo: "rater" → "rather" (Page 13).
2. Consider evaluating newer models (e.g., o3-mini) to strengthen claims like scalability and token efficiency.

**Other Strengths And Weaknesses:**

None

**Questions For Authors:**

Could the authors expand their analysis to include newer models (e.g., o3-mini high/medium/low, Gemini 2.0 Flash Thinking, Grok-3, and QwQ-32B) to provide additional data points for validating the curse of complexity and hidden reasoning token dynamics? Such comparisons would strengthen the generalizability of the findings, particularly for scaling trends and hidden token analysis.

**Relation To Broader Scientific Literature:**

The work aligns with prior studies on LLM benchmarking, especially for reasoning (like MATH).

**Theoretical Claims:**

The paper focuses on empirical findings.

---

> ### Author Rebuttal · Authors · 2025-04-01
>
> Thank you very much for your thoughtful review! We address each of the concerns and questions raised, and we will incorporate these clarifications and additional analyses into revision of the paper. We hope our responses adequately resolve the issues highlighted and kindly ask for your reconsideration of the scores if appropriate.
>
> #### W1: Lack of evaluation on the latest models like o3-mini
> Thank you very much for the suggestion! The O3-mini APIs were not stable at the time of our experiments, which often leads to timed out errors and refusals such as "Invalid prompt: your prompt was flagged as potentially violating our usage policy." The O3-mini-high model was particularly unstable.
> We recently re-ran the experiments and will include the results in the  version. The current results are as follows:
>
> | MODEL                  | XL   | Large | Medium | Small | Cell Acc |
> |------------------------|------|-------|--------|-------|----------|
> | o3-mini-2025-01-31-high | 75.5 | 87.5  | 97.1   | 99.7  | 95.7    |
> | o1-2024-12-17          | 42.5 | 78    | 92.1   | 97.2  | 78.7     |
> | deepseek-R1            | 28.5 | 73.5  | 95.7   | 98.4  | 80.5     |
> | o3-mini-2025-01-31-low | 23   | 64.5  | 91.1   | 99.4  | 72.6     |
> | o1-preview-2024-09-12  | 17   | 59.5  | 88.2   | 98.1  | 75.1     |
> | o1-mini-2024-09-12     | 12   | 39    | 76.8   | 87.5  | 70.3     |
>
> We will include the results in the camera-ready version.
>
> #### Q1: Could the authors expand their analysis to include newer models (e.g., o3-mini high/medium/low, Gemini 2.0 Flash Thinking, Grok-3, and QwQ-32B) to provide additional data points for validating the curse of complexity and hidden reasoning token dynamics? Such comparisons would strengthen the generalizability of the findings, particularly for scaling trends and hidden token analysis.
>
> Yes, we are more than happy to expand the analysis to include newer models and do more analysis with their reasoning process, and find the common failure modes for suggesting the future direction of the research. Please satey tuned for our leaderboard website for the latest results and an easy-to-use tool for evaluating all LLMs.
>
>
> #### Typo: "rater" → "rather"
>
> Thank you for pointing out! We will fix it in the camera-ready version, and will carefully proofread the paper to avoid similar typos.
>
> ---
>
> Thank you so much for your review! We will address your suggestions in the  version. We hope these clarifications and additional analyses adequately resolve the issues highlighted.

---

### Official Review · Reviewer_zqZW · 2025-03-16

**Overall Recommendation:** 3

**Summary:**

This paper investigates the logical reasoning capabilities of large language models (LLMs) by introducing ZebraLogic, a benchmark dataset of 1,000 logic grid puzzles. These puzzles are formulated as constraint satisfaction problems (CSPs) with controlled complexity levels, allowing for systematic evaluation of LLMs' reasoning abilities across varying difficulties. The authors use two key metrics to measure puzzle complexity: search space size and Z3 conflict count (the number of conflicts encountered by an SMT solver). The research also describes what the authors call the "curse of complexity," a considerable loss in LLM performance as issue complexity grows, even when expanding to bigger models.

**Claims And Evidence:**

1. **Performance decline with complexity:** The authors demonstrate this through comprehensive evaluations across various model sizes and architectures, showing consistent performance drops as complexity increases.
2. **Limitations of model scaling:** The experiments show that even the largest models (e.g., Llama-3.1-405B) achieve near-zero accuracy on highly complex puzzles, supporting the claim that model scaling alone cannot overcome reasoning limitations.
3. **Optimal reasoning token ratio:** The claim that there exists an optimal ratio of reasoning tokens to Z3 conflicts is supported by their analysis of o1 models, though this evidence is more correlational than causal.

**Essential References Not Discussed:**

N/A

**Experimental Designs Or Analyses:**

1. The authors evaluate a diverse set of models spanning different architectures, sizes, and both open-weight and proprietary systems.
2. One potential limitation is that the analysis of o1's hidden reasoning tokens relies on estimates since these tokens aren't directly accessible. However, the authors acknowledge this limitation and verify their estimates once o1-full is released.
3. The Best-of-N sampling experiments are insightful, though the choice of majority voting and reward models for candidate selection could be expanded to explore other selection strategies.

**Methods And Evaluation Criteria:**

1. The two complexity metrics (search space size and Z3 conflicts) provide complementary views of problem difficulty.
2. The categorization of puzzles into four complexity groups enables a clear analysis of performance trends.
3. The evaluation across multiple model architectures and sizes allows for robust comparative analysis.
4. The puzzle generation methodology using clue types and templates is sound, ensuring puzzles have unique solutions while maintaining varied difficulty levels.

**Other Comments Or Suggestions:**

1. A more detailed discussion of how these findings might inform training objectives for future models would strengthen the paper.

**Other Strengths And Weaknesses:**

**Strengths:**
1. The multi-dimensional analysis of scaling behavior (model size, sampling, reasoning tokens).
2. The inclusion of both open-weight and proprietary models for comprehensive evaluation.

**Weaknesses:**
1. The analysis of o1's reasoning process relies on limited visibility into its hidden reasoning tokens
2. The self-verification experiments could be expanded to explore more sophisticated reflection mechanisms
3. The paper doesn't extensively analyze what specific types of reasoning failures occur as complexity increases

**Questions For Authors:**

1. The paper shows that o1 models generate ~10x more hidden reasoning tokens than standard models. Have you explored whether standard models could benefit from being allowed to generate similarly extensive reasoning steps, or are there architectural differences that make this approach uniquely effective for o1?
2. How might the findings from ZebraLogic generalize to other formal reasoning domains beyond logic grid puzzles? Do you expect similar complexity scaling behaviors for tasks like mathematical reasoning or program synthesis?
3. Have you analyzed what specific types of reasoning errors or failure modes emerge as puzzle complexity increases? This could provide insights into targeted interventions for improving reasoning capabilities.

**Relation To Broader Scientific Literature:**

The paper's core contribution of showing how performance scales across multiple dimensions (model size, sampling, reasoning tokens) provides a more comprehensive picture. The paper builds on prior work on logical reasoning benchmarks like LogiQA [1] and related investigations of LLM reasoning limits. It extends previous research on grid puzzles [2, 3, 4] by systematically controlling for complexity.

[1] Logiqa: A challenge dataset for machine reading comprehension with logical reasoning. Liu et. al, IJCAI 2020.
[2] Learning to Automatically Solve Logic Grid Puzzles, Mitra et al, EMNLP 2015
[3] Faith and Fate: Limits of Transformers on Compositionality, Dziri et al, NeurIPS 2023.
[4] Step-by-Step Reasoning to Solve Grid Puzzles: Where do LLMs Falter?, Tyagi et al, EMNLP 2024.

**Theoretical Claims:**

There are no theoretical proofs to verify in this paper. The authors establish that ZebraLogic is NP-complete through reduction from the Quasigroup Completion Problem. The paper primarily focuses on empirical evaluation rather than theoretical derivations.

---

> ### Author Rebuttal · Authors · 2025-04-01
>
> Thank you very much for your thoughtful review! We address each of the concerns and questions raised, and we will incorporate these clarifications and additional analyses into revision of the paper. We hope our responses adequately resolve the issues highlighted and kindly ask for your reconsideration of the scores if appropriate.
>
> #### W1: The analysis of o1's reasoning process relies on limited visibility into its hidden reasoning tokens
>
> We recognize that OpenAI's restrictions on raw reasoning tokens limited our o1 analysis. To gain deeper insights, we will analyze DeepSeek's R1 model's visible reasoning tokens in the revised paper.
>
> #### W2: The self-verification experiments could be expanded to explore more sophisticated reflection mechanisms
>
> Thank you for this insightful suggestion. We recognize the value of exploring more advanced reflection mechanisms to enhance our analysis. To address this, we will conduct a detailed investigation into sophisticated reflection approaches and incorporate the findings into the  version of the paper, ensuring a more comprehensive evaluation of their impact.
>
> #### W3 & Q3: The paper doesn't extensively analyze what specific types of reasoning failures occur as complexity increases
>
> Thank you for your valuable suggestion regarding the analysis of reasoning failure modes. We acknowledge that the page constraints of the initial submission limited our ability to include a comprehensive discussion on this topic. To address this concern thoroughly in the revised paper, we will incorporate a detailed examination of specific failure modes, including:
>
> - A quantitative breakdown of the number and types of constraints (e.g., uniqueness, clue-based, positional) that remain unsatisfied in LLM outputs, identifying prevalent error patterns such as failures in handling non-monotonic reasoning or spatial constraints as complexity scales.
> - An examination of how the sequence and presentation of clues influence solution accuracy, testing for systematic biases or dependencies in the reasoning process, such as over-reliance on early clues or misinterpretation of later ones.
> - Case studies from our human evaluation (Appendix C.1) featuring specific examples of reasoning breakdowns, such as incomplete backtracking or incorrect counterfactual assumptions, particularly in puzzles with large search spaces or high Z3 conflict counts.
>
> Our analysis of the o1 outputs reveals two predominant failure modes that significantly impact its reasoning performance:
> 1) **Lazy Mode (Most Prevalent):** The model often uses brief summaries instead of detailed step-by-step reasoning (e.g., "going step-by-step" or "cycling through possibilities"), reducing solution robustness.
> 2) **Incorrect Proof of Impossibility:** The model wrongly claims clues are unsatisfiable, misinterpreting constraints like adjacency, leading to premature, incorrect conclusions and exposing limitations in complex constraint handling.
>
> Thanks again for your valuable suggestion! We will address this in revision.
>
> #### Q1: The paper shows that o1 models generate ~10x more hidden reasoning tokens than standard models....
>
> Thanks for the great question! We think the model architecture (i.e., the number of parameters, the number of layers, the number of attention heads, etc.) is not a key factor but still the data distribution and training methods are important. DeepSeek's R1 vs V3 is a good example for this. Training the base model with longer CoT data for triggering self-verification and self-correction is what enables R1 to outperform V3 on the ZebraLogic benchmark. Recent paper [1] also suggests that keeping appending the token "Wait" to force the model to think more is effective for improving the reasoning performance of the model.
>
> [1] s1: Simple test-time scaling
>
>
> #### Q2: How might the findings from ZebraLogic generalize to other formal reasoning domains beyond logic grid puzzles? Do you expect similar complexity scaling behaviors for tasks like mathematical reasoning or program synthesis?
>
> Yes, we believe ZebraLogic's findings apply to other formal reasoning domains like mathematical reasoning and code generation, as many reasoning challenges are constraint satisfaction problems, central to our research. Although quantifying complexity in math or coding is harder than in our grid size and Z3 conflict measures, constraint-based reasoning principles suggest strong generalization potential.
>
> #### Q3: Have you analyzed what specific types of reasoning errors or failure modes emerge as puzzle complexity increases?
>
> Please refer to the answer for W3 for this question.
>
>
>
>
> ---
>
> Thank you for your review! We will address your suggestions in the  version. We hope these clarifications and additional analyses adequately resolve the issues highlighted and kindly ask for your reconsideration of the scores if appropriate.

---

### Decision · Program_Chairs · 2025-05-01

**Decision:**

Accept (poster)

**Comment:**

The paper presents a synthetic Logical benchmark with increasing complexity. Reviewers appreciated the insights on language models' accuracy as the problem complexity is increased and the limitations of model scaling. The paper also studies the effects of scaling test-time compute and how it may not be enough to solve the hardest problems. The evaluation is comprehensive and the results useful for the reasoning community, with the inclusion of both open-weight and proprietary models.
The paper is timely and presents a useful benchmark.